# Iron and sulfide nanoparticle formation and transport in nascent hydrothermal vent plumes

Alyssa J. Findlay [1,2], Emily R. Estes [3], Amy Gartman[4], Mustafa Yücel[5], Alexey Kamyshny Jr.[2] & George W. Luther III [3]

Deep-sea hydrothermal vents are a significant source of dissolved metals to the global oceans, producing midwater plumes enriched in metals that are transported thousands of kilometers from the vent source. Particle precipitation upon emission of hydrothermal fluids controls metal speciation and the magnitude of metal export. Here, we document metal sulfide particles, including pyrite nanoparticles, within the first meter of buoyant plumes from three high-temperature vents at the East Pacific Rise. We observe a zone of particle settling 10–20 cm from the orifice, indicated by stable sulfur isotopes; however, we also demonstrate that nanoparticulate pyrite ($FeS_2$) is not removed from the plume and can account for over half of the filtered Fe ($\leq$0.2 µm) up to one meter from the vent orifice. The persistence of nanoparticulate pyrite demonstrates that it is an important mechanism for near-vent Fe stabilisation and highlights the potential role of nanoparticles in element transport.

[1] Center for Geomicrobiology, Department of Bioscience, Aarhus University, Aarhus C 8000, Denmark. [2] Department of Geological and Environmental Sciences, Ben-Gurion University of the Negev, Beer Sheva 84105, Israel. [3] School of Marine Science and Policy, University of Delaware, Lewes 19958 DE, USA. [4] U.S. Geological Survey, P.C.M.S.C., 2885 Mission St., Santa Cruz 95060 CA, USA. [5] Institute of Marine Sciences, Middle East Technical University, 33731 Mersin, Turkey. Correspondence and requests for materials should be addressed to A.J.F. (email: afindlay@bios.au.dk)

The fluids emitted from most deep-sea hydrothermal black smoker vents are enriched in iron, other metals and sulfide up to several orders of magnitude over ambient seawater. Many of these elements are biogeochemically relevant; for example Fe sourced from vents is a globally important component of the oceanic Fe budget[1,2], and thus may ultimately affect primary production in the surface ocean[3,4]. Quantifying the significance of element transport from hydrothermal vents is therefore of broad oceanographic importance.

Determining the potential for element transport away from the vent source is complicated because the vent fluid undergoes a wide spectrum of biogeochemical changes between emission of the fluid and the formation of midwater plumes. In particular, mixing of the reduced, hot and acidic vent fluid with cold, oxygenated seawater leads to the precipitation of many metals in the area surrounding the vents as sulfide or oxide minerals[5,6]. The formation, aggregation, oxidation and/or stabilisation of particles and nanoparticles in emitted vent fluid directly determines the potential for element transport to the broader oceans. Moreover, although the chemistry of end-member vent fluid and nascent buoyant plumes is highly variable between vents[7–9], the chemistry of the non-buoyant plume appears to be consistent, regardless of the initial fluid composition[10]. This consistency indicates that it is the physical and (bio)geochemical processes in the initial stages of the buoyant plume that exert the primary control on which elements are exported from vents, and that these processes are similar between different black smoker vent fields. In particular, the speciation and solubility of reduced Fe and S are extremely sensitive to oxidation along the steep redox gradients found at the vent orifice, which may result in their loss from the rising fluid. The dynamics of reduced Fe and S are important as these elements dominate fluid chemistry in many systems and therefore affect the speciation of other elements, such as Pb, Co, Cd, Cu and Zn[9,11].

Nevertheless, recent work has demonstrated that Fe may be stabilised as nanoparticulate pyrite in the hydrothermal fluid[12,13] and by organic ligands within the colder buoyant and non-buoyant plume[14–16] and that this Fe is transported hundreds to thousands of kilometers from the local vent area[17–20]. These findings raise important questions regarding the processes affecting particle formation and transport in the initial mixing zone, including the extent of metal sulfide precipitation and settling. However, systematic studies of particle formation and Fe dynamics within this initial mixing zone are lacking.

In order to elucidate these processes and investigate the potential for Fe transport from the vent sites, we determined the speciation of Fe and S and quantified the abundance of nanoparticulate pyrite within the first meter of buoyant plumes from three hydrothermal vents with high sulphide to iron ratios (ranging from 4 to 10) at the fast-spreading East Pacific Rise (EPR, 9°50′N); P Vent, Bio 9 Vent and Bio Vent. We show that despite a decrease in total Fe and the formation and entrainment of sulfide particles within the plume, nanoparticulate pyrite can comprise up to 60 % of filtered Fe (≤0.2 μm) one meter from the vent orifice, even though it is not detectable at statistically significant concentrations at the point of emission by chemical leaching methods. This, combined with evidence for settling of sulfide particles in the plume from stable sulfur isotope measurements, indicates that physical and geochemical controls on sulfide particle formation and settling have broad implications for the stabilisation of Fe in nanoparticulate pyrite and for metal transport from hydrothermal vents.

## Results and Discussion

**Fe and S dynamics in the first meter of the buoyant plume.** The concentration and speciation of iron and sulphur were measured in samples from three different black smoker vents up to one meter from the vent orifice (Supplementary Figure 1). Within this distance, Fe was predominately (≥80 %) in the ≤0.2 μm fraction (Supplementary Table 1). Despite this, total unfiltered Fe was exponentially removed at P Vent and Bio 9 Vent, particularly in the first 10 centimeters of the buoyant plume, indicating the formation and subsequent rapid settling of larger particles. At Bio Vent, initial Fe concentrations were lower and there was much less Fe removal within the initial stages of the plume (Fig. 1a, Supplementary Table 1).

Total unfiltered Acid Volatile Sulfide (AVS), which comprises free sulfide and less recalcitrant metal sulfides (e.g. FeS, ZnS), demonstrates behaviour consistent with removal within the first 10 centimeters at all three vents (Fig. 1b) and AVS can account for the observed Fe removal. Concentrations of total Chromium Reducible Sulphur (CRS), consisting of more recalcitrant sulfur minerals[21] (e.g. $FeS_2$, $FeCuS_2$ and partially $S^0$), are up to an order

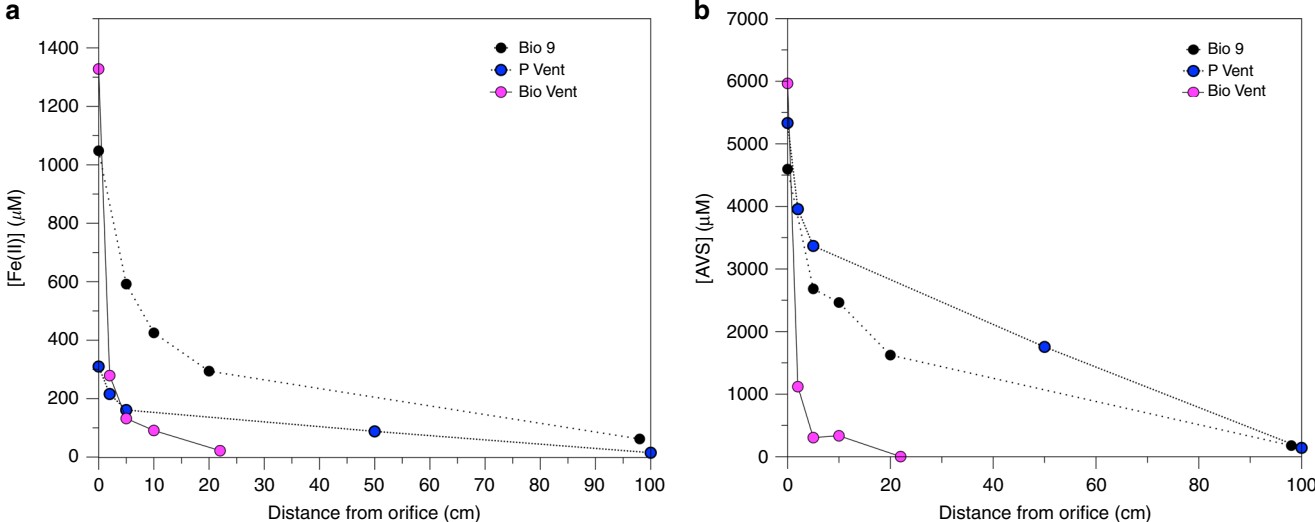

**Fig. 1** Plume iron and sulphur geochemistry. **a** Total unfiltered Fe(II) concentrations in all three plumes and **b** Total unfiltered AVS concentrations in all three plumes

**Table 1 Nanoparticulate pyrite and elemental sulfur concentrations in each sample**

| Vent | T | pH | [Mg] (mM) | Distance from orifice (cm) | [Nano py] (μM) | % $Fe_{npy}$/ $Fe_{total}$ ≤0.2 μm | [$S^0$]<0.2 (μM) | [$S^0$]>0.2 (μM) | [$S^0$] total (μM) |
|---|---|---|---|---|---|---|---|---|---|
| Bio 9 | 373 | 3.13 | 3.40 | 0 | BDL | — | 30 | 2.3 | 32 |
| | 150 | 4.87 | 43.7 | 10 | 43 | 20 | 0.99 | n/a | 0.99 |
| | 79 | 5.31 | 52.2 | 20 | 24 | 20 | BDL | 1.9 | 1.9 |
| | 25 | 5.58 | 51.4 | 50 | 9 | 12 | BDL | 18 | 18 |
| | 10 | 6.28 | 51.1 | 100 | 1 | 6 | 1.3 | 22 | 23 |
| P Vent | 350 | 3.27 | 5.30 | 0 | BDL | — | 2.4 | 1.7 | 4.2 |
| | 240 | 3.93 | 25.9 | 5 | 20 | 4 | 3.2 | 18 | 22 |
| | 210 | 4.01 | 28.3 | 10 | 18 | 6 | 0.41 | 30 | 30 |
| | 120 | 4.87 | 46.0 | 50 | 13 | 6 | 0.62 | 50 | 50 |
| | 35 | 5.73 | 51.5 | 100 | 16 | 30 | 3.8 | 28 | 32 |
| Bio Vent | 310 | 3.77 | 11.5 | 0 | BDL | — | 4.3 | 10 | 15 |
| | 260 | 4.10 | 20.4 | 20 | BDL | — | 1.9 | 17 | 19 |
| | 160 | 4.34 | 32.4 | 30 | BDL | — | BDL | 17 | 17 |
| | 60 | 4.77 | 47.1 | 50 | BDL | — | BDL | 17 | 17 |
| | 10 | 5.79 | 54.0 | 100 | 9 | 60 | BDL | 0.96 | 0.96 |

pH values are shipboard measurements
BDL, below detection limit (nano pyrite 1 μM: $S^0$ 0.5 μM)

of magnitude lower than those of AVS and consist of 1–13 % of the total sulphide (Fig. 1b, Supplementary Table 1).

Within the initial first meter of mixing, total unfiltered Fe remains almost exclusively as Fe(II) (Supplementary Table 1). The predominance of Fe(II) is consistent with previous studies which showed that Fe remains reduced at least up to 1.5 m from the vent orifice at the Mid-Atlantic Ridge[9,13]. Although oxygenated ocean water mixes into the buoyant plume within the first meter with seawater (Table 1), the excess sulphide with respect to Fe present in these fluids keeps Fe reduced due to a catalytic cycle (Eq. 1a, b).

$$4Fe^{2+} + O_2 + 4H^+ \rightarrow 4Fe^{3+} + 2H_2O \qquad (1a)$$

$$2\,Fe^{3+} + H_2S \rightarrow 2Fe^{2+} + S^0 + 2H^+ \qquad (1b)$$

In addition to the persistence of Fe(II), the presence of filterable and particulate $S^0$ within the plume (Table 1) provides further evidence for such a catalytic cycle. Generally, the particulate fraction of $S^0$ is lowest in the samples taken from the orifice, whereas the <0.2 μm fraction is greatest in these samples. This indicates that $S^0$ is mainly formed from sulfide oxidation in the plume and is not due to entrainment of chimney particles. The $S^0$ in the < 0.2 μm fraction in the end-member samples is expected to be dissolved or nanoparticulate $S^0$, as polysulfides undergo hydrolysis rapidly at high temperature ( ≥150 °C)[22]. The decrease in $S^0$ in the < 0.2 μm fraction is likely due to changes in its solubility over the temperature gradient in the initial plume[23].

**Formation of pyrite and persistence of pyrite nanoparticles.** Significant concentrations of nanoparticulate pyrite (defined operationally as the difference between nitric acid and HCl soluble Fe in the ≤0.2 μm filtered fraction[12,13]) were measured in all plume samples at Bio 9 and P Vent (Table 1; Supplementary Figure 2). In contrast, pyrite nanoparticles were not statistically detectable in the hot (>350 °C) end-member samples by this leaching method. Pyrite was detected in bulk end-member and plume samples by powder XRD at both Bio 9 and P-Vent, respectively, but was not detected at Bio Vent (Supplementary Table 2).

Evidence for bulk pyrite formation comes from the stable isotope fractionation between total unfiltered AVS and CRS at P

Vent and Bio 9, which are consistent with partial isotopic equilibrium between pyrite and $H_2S$ through an FeS precursor[24] (Eq. 2; Fig. 2).

$$FeS + H_2S \rightarrow FeS_2 + H_2 \qquad (2)$$

The FeS−$H_2S$ pathway is expected to be the predominant mechanism for pyrite formation in buoyant vent plumes based on pH, reactant availability and reaction kinetics[25]. The rate of the reaction presented in Eq. 2 plateaus above 125 °C as the reaction is diffusion controlled[26] and decreases with decreasing $H_2S$ concentration and at temperatures lower than 125 °C. Thus, the most significant pyrite formation should occur near the vent orifice, and the nanoparticulate pyrite detected higher in the plume has likely persisted throughout the cooling and rise of the higher temperature fluid. At high temperatures, nucleation rates are high, favouring the formation of nanoparticles[27], and the rapidly cooling temperatures and decreasing reactant concentrations during mixing limit further growth, and thereby settling, of the nanoparticles. Interestingly, in contrast to the chimney pyrite from Bio 9 analysed by Rouxel et al.[28], the S isotope composition for the total unfiltered fluid samples measured here from Bio 9 and P Vent fall mostly below the pyrite-$H_2S$ equilibration line. This may be due to formation of pyrite (e.g., CRS) at high temperatures, followed by removal of light $H_2S$ from the solution, either due to metal sulphide precipitation or degassing of $H_2S$, both of which would leave the remaining AVS pool isotopically heavy.

In contrast to Bio 9 and P Vent, statistically significant concentrations of nanoparticulate pyrite were only observed in one sample at Bio Vent (Table 1), one meter from the orifice in the buoyant plume. Pyrite was also not detected in bulk samples via XRD (Supplementary Table 2). The apparent isotopic fractionation between AVS and CRS falls directly on the FeS−$H_2S$ equilibration line[29], indicating that metal monosulphides, rather than $FeS_2$ are the major particles forming at Bio Vent due to the lower Fe concentrations (Fig. 1a; Supplementary Table 1) and cooler temperature of the emitted fluid (i.e. 310 °C vs. ≥350 °C). Importantly, at all vent sites, the distribution of nanoparticulate pyrite in the plume (Table 1) does not indicate that it is removed within the first meter. At P vent, for example, concentrations of nanoparticulate pyrite are nearly constant as temperature decreases.

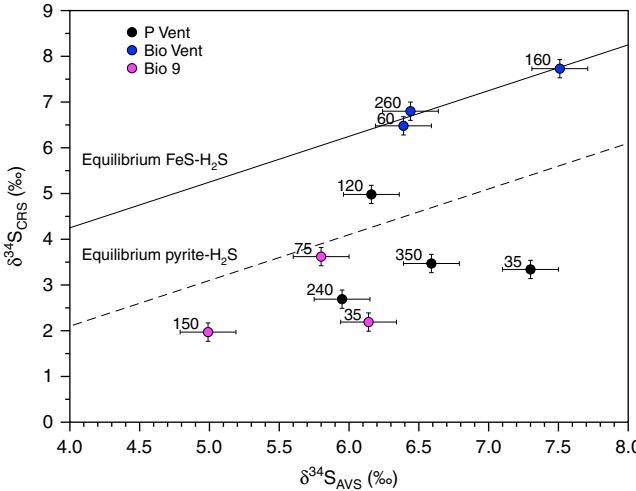

**Fig. 2** Sulphur isotopic composition of AVS and CRS in the buoyant plume. Paired $\delta^{34}S_{AVS}$ and $\delta^{34}S_{CRS}$ values from plume samples showing the expected values based upon isotopic equilibration between FeS–H$_2$S and pyrite–H$_2$S (the experimental equilibrium values were taken from Syverson et al.[24] and were determined for T = 350 °C. We note that it is not certain what the isotope fractionation is at lower temperatures). Note that CRS values are not available for all AVS samples from Fig. 2 due to small sample size. Labels next to the data points represent the in situ temperature (°C) at which the samples were taken

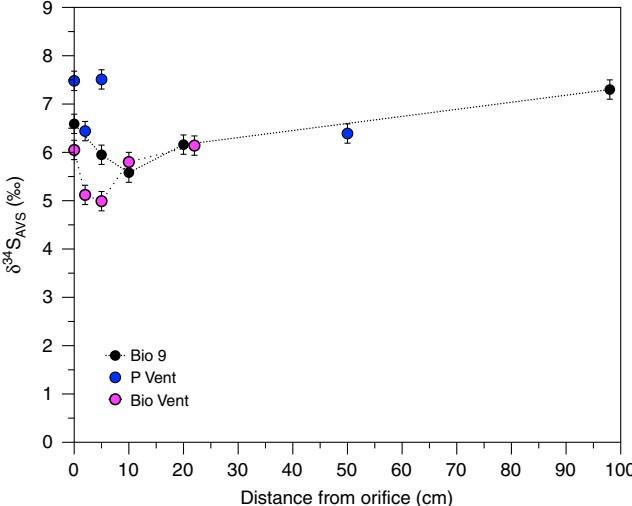

**Fig. 3** Sulphide isotopic composition in the buoyant plume. $\delta^{34}S_{AVS}$ (unfiltered) with increasing distance from the vent orifice in the plume. Error bars represent one standard deviation

**Sulphide precipitation and implications for element export.** The value of total unfiltered $\delta^{34}S_{AVS}$ over the plume for P Vent and Bio 9 decreases and then increases again with increasing distance from the vent orifice (Fig. 3). These isotopic differences are larger than the typical variability for end-member samples[30,31]. This pattern is not consistent with fractionation caused by reaction or particle loss from the plume due to mass balance considerations; however, it does correspond to the portion of particulate Fe, as well as the visual particle density in the samples (Supplementary Figure 3, Supplementary Table 1). Although the isotope fractionation between sulfide and FeS is small ($\leq 0.25^0/_{00}$), there is a fractionation of $2.7^0/_{00}$ between dissolved H$_2$S and HS$^-$, by which HS$^-$ is lighter[32]. FeS and other metal sulfides within the plume are expected to be formed by reaction of metals with HS$^-$, rather than H$_2$S as the pH increases. The pK$_1$ of H$_2$S in seawater at 150 °C is 6.08[33], so that the pH of the fluids after 10–20 cm begins to approach the pK$_1$. Furthermore, in situ pH is typically higher than the shipboard pH; the in situ pH in the end-member fluid at P Vent was previously measured to be about 5.5[34], compared with our shipboard measurement of 3.27 (Table 1).

Metal sulfide particles should therefore be isotopically lighter than coexisting H$_2$S. If the distribution of these particles were homogenous throughout the plume, there would be no apparent difference in $\delta^{34}S_{AVS}$, but both visual (Supplementary Figure 3) and Fe distribution data (Supplementary Table 1) demonstrate that particle distribution is not homogenous. Therefore, the more depleted $\delta^{34}S_{AVS}$ values must reflect a concentration of particles (and thereby isotopically light sulfide) within the plume within the first 10–20 cm at intermediate temperatures (100–250 °C). Although higher particle concentrations are due in part to the increased amount of particle formation at cooler temperatures and higher pH[35], the $\delta^{34}S_{AVS}$ patterns can only be explained if bulk particles are concentrated in this area by settling from higher in the plume. Particle settling is induced as the particles grow with distance from the orifice. Simultaneously, the upward flow decreases, changing the buoyancy (Fig. 4). This observation

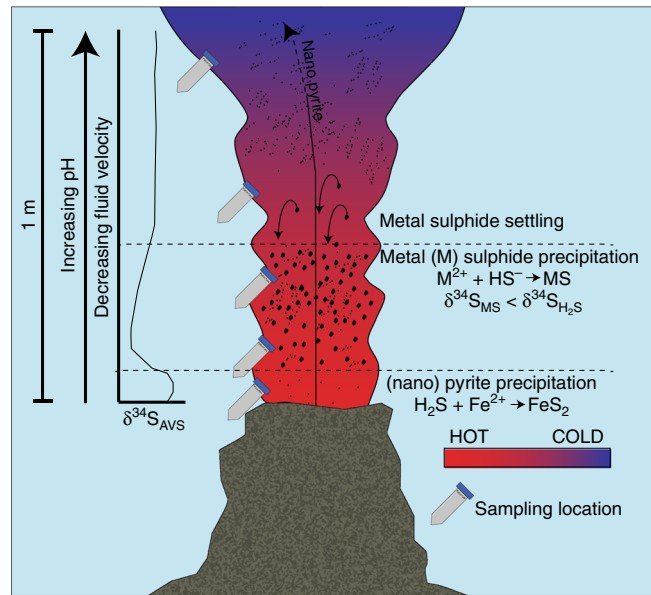

**Fig. 4** Particle settling and isotope effects in the buoyant plume. Conceptual model of particle settling within the initial buoyant plume and the resulting isotopic effects

indicates a critical point within the nascent plume that is decisive for the quantity and type of particles and elements that will be transported further up in the water column to the neutrally buoyant plume, where they can then be transported laterally.

The location and extent of this zone will be affected by the physical characteristics of a given vent, including the temperature of the emitted fluids and the size and flow rate of the plume. For example, the isotopic composition of AVS in the Bio Vent plume does not follow the same trend observed at Bio 9 and P Vent. This is consistent with the visually lower flow and particle concentration in this plume, as well as the observed lower removal of Fe and higher ratio of sulfide to Fe, compared to Bio 9 and P Vent (Supplementary Table 1). These characteristics suggest moreover that major Fe and particle removal took place at an earlier point along the fluid flow pathway, for example within the chimney.

For all vents, nanoparticulate pyrite is a significant portion (up to 60%) of ≤0.2 μm Fe at 1 m above the vent orifice and above this particle concentration zone. The lack of evidence for removal of nanoparticulate pyrite in any of the three plumes, despite settling of bulk particles, indicates that pyrite nanoparticles stabilise Fe with respect to particle growth, within the ≤0.2 μm filtered fraction, limiting the extent of Fe that settles from the plume. Due to their small size[12] and resistence to oxidation[36], pyrite nanoparticles may thus mediate Fe export from the vents to the global oceans.

Previous work has shown the widespread presence of $FeS_2$ nanoparticles in the end-member fluid of a variety of different hydrothermal settings with different geological (back arc basin, slow and fast spreading centers) and chemical characteristics (Fe to sulphide ratios ranging from 0.1 to 10)[12,13]. Here, using a combination of chemical leaching, mineralogical and stable isotope methods, we demonstrated that these nanoparticles serve to stabilise Fe during the extensive mixing and particle precipitation that occurs immediately upon fluid emission from the vent orifice. This study shows that particle formation, settling and buoyancy between the hot vent fluids and the neutrally buoyant plume directly affect metal transport from the initial mixing zone. The formation of nanoparticulate pyrite at high temperatures and limits on its growth due to rapidly cooling temperatures during mixing act to stabilise the hydrothermally derived Fe as nanoparticulate pyrite and prevent further particle growth and oxidation. The data presented here demonstrate for the first time the persistence of nanoparticulate pyrite throughout a zone of major particle growth and settling, and suggest the possibility for continued nanoparticle transport, including transport from the vent sites.

## Methodology

**Sampling**. Samples were taken using the Human Operated Vehicle Alvin II operated by the RV *Atlantis* at three different vent sites (P Vent, Bio 9 and Bio Vent) along the East Pacific Rise 9°N during March and April 2017. The in situ temperature of each sample was taken with the high-temperature probe attached to the titanium Major Sampler. One sample was taken within the vent orifice, then subsequent samples were taken from first meter of the rising plume in order to capture a broad temperature gradient. Great care was taken by the Alvin pilots to hold the nozzle of the sampler in the centre of the plume during sampling.

**Sample processing**. The samples were processed immediately upon recovery shipboard, typically 2–4 h after they were taken. The shipboard pH and temperature were measured immediately and samples were fixed for either shipboard or shorebased analyses.

**Fe concentration and speciation**. For Fe samples, triplicate aliquots of 10 mL (0.2 μm filtered and unfiltered) were added to centrifuge tubes containing 0.5 mL HCl (trace metal grade). Iron measurements were performed after sequential leaching with HCl and $HNO_3$[12]. Each acid leach was allowed to react for at least 8 hours. Spectrophotometric measurements were made onboard the ship using the ferrozine method of Stookey[37] at a wavelength of 562 nm. Samples were buffered in 2.5 M ammonium acetate prior to addition of the ferrozine reagent. After the initial analysis of the HCl leach, the samples were treated with hydroxylamine hydrochloride as a reducing agent to measure total iron. Fe(III) was determined by the difference between these numbers. After the $HNO_3$ leach, hydroxylamine was added simultaneously with the ferrozine reagent to measure total Fe. Pyrite concentrations were determined by subtracting total Fe from the sum of Fe(II)

and Fe(III) determined after the HCl leach. The detection limit of this method was 100 nM.

**Sulphide speciation and concentration**. Samples for sulfide analysis (AVS/CRS) were preserved by adding 2 mL NaOH (0.5 M) and 2 mL zinc acetate (0.1 M) to 2 mL of sample, which was then frozen and stored at −20 °C. Measurements were made on shore using the distillation method of Fossing and Jørgensen[21]. Samples preserved in NaOH and zinc acetate were defrosted and the entire sample was transferred to a bubbler system along with 12 mL Ar-purged MQ water. In the bubbler, samples were purged with Ar for an additional 10 min. 0.5 mL of 3 M HCl (trace metal grade) was then added via syringe to the system and evolved $H_2S$ was captured in 20 mL 1 M trace metal clean NaOH. The base deprotonates the $H_2S$ gas, converting it to $HS^-$ and allowing direct quantification via UV-Vis spectroscopy ($HS^-$ peak at 230 nm). Reaction time prior to analysis was 1.5 hours. This initial measurement represents the Acid Volatile Sulfide fraction. Test tubes containing a fresh 20 mL aliquot of 1 M trace metal clean NaOH were then placed in-line with the bubbler system and 0.5 mL of 1 M Cr(II) solution in 1 M HCl (prepared using a Jones reduction column) was injected into the sample via syringe. Samples were again bubbled for 1.5 h prior to analysis of evolved $H_2S$ as $HS^-$ trapped in NaOH. This second analysis represents the concentration of Chromium-Reducible Sulfide.

**Elemental sulphur**. Sample aliquots of 30–50 mL were filtered through a 0.2 μm Millipore GTTP filter into a zinc acetate solution to fix free sulfide and prevent oxidation during the extraction. The filtrate was then extracted shipboard in 5 mL of toluene for 1.5 h[38]. The toluene layer was separated and stored at −20 °C for later analysis. The filters were also stored at −20 °C for later extraction and analysis onshore. Elemental sulfur was quantified by HPLC using a C-18 column and 98% methanol 2% water as the eluent with UV detection at 230 nm. The retention time for $S_8$ was approximately six minutes. The detection limit for this method is 0.5 μM.

**Stable sulfur isotope measurements**. Unfiltered 5–50 mL samples fixed in zinc acetate were distilled via the two-step AVS/CRS procedure outlined above to separate AVS and CRS. Evolved sulfide was trapped as ZnS and was converted to $Ag_2S$ upon addition of $AgNO_3$ (1 M). The precipitate was aged one week, washed with 18.2 MΩ water (MilliQ) and $NH_4OH$ (1 M), then dried overnight at 60 °C. $Ag_2S$ was converted to $SF_6$ by reaction with excess $F_2$ at 300 °C for at least 10 hours in Ni alloy reaction chambers. The $SF_6$ was then purified cryogenically and by preparative gas chromatography. Following purification, stable sulphur isotopic measurements were conducted on a Finnigan MAT 253 dual inlet mass spectrometer[39].

Isotopic composition is presented in permil using standard δ notation relative to VCDT (Eq. 3)

$$\delta^{34}S = {}^{34}R_{sample}/{}^{34}R_{VCDT} - 1 \qquad (3)$$

in which $^{34}R = {}^{34}S/{}^{32}S$.

**XRD**. Samples for XRD analysis were prepared by the centrifugation of 300–700 mL of unfiltered fluid[13]. Samples were then capped with UHP nitrogen and frozen. For analysis, pelleted samples were resuspended in Milleq ® water and evaporated under nitrogen onto zero diffraction wafers by MTI. Samples were run on a Panalytical X'Pert3 Powder XRD using a Cu Kα source at 40 kV and 45 mA, and scanned from 5–70° 2θ. Three scans were performed with a 0, −1, and + 1 degree wobble in order to avoid preferential orientation; final scans were an

average of all three scans. HighScore Plus software was used for peak identification and fitting, with International Center for Diffraction Data 2014 used for sample ID. Mineralogy as determined from XRD was checked for compatibility with elemental results obtained from SEM/EDS. SEM was performed using a Tescan VP-SEM in high vacuum mode at an accelerating potential of 20 kV.

**Mg concentrations**. Samples for Mg concentrations were filtered shipboard (0.2 μm) and acidified in 0.75 M nitric acid. Onshore, samples were diluted between 1:100 and 1:2000, using sample temperature as an estimate of Mg. Samples were analyzed on a Thermo Electron Corporation Finnigan Element XR Inductively Coupled Plasma Mass Spectrometer (ICP-MS) in low resolution mode, using a rhodium internal standard. Dilution preparation and sample analysis were conducted under trace metal clean conditions.

## Data Availability
All data generated or analysed during this study are included in this published article (and its supplementary information files).

## Code Availability
No custom code was used in the production or analysis of these data.

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

## Acknowledgements
This work was supported by a grant from the Marine Geology and Geophysics Program of the U.S. National Science Foundation (OCE-1558712) to G.W.L. and both a Kreitman Postdoctoral Fellowship (BGU) and a Marie Curie Fellowship to A.J.F. (MSCA Sed-SulphOx 746872). We thank the DSV *Alvin* group and the Captain and crew of the R/V *Atlantis* for their excellent work. André Pellerin is thanked for assistance with Fig. 4.

## Author contributions
A.J.F. conceived the study, participated in sampling, conducted analyses, interpreted the data and wrote the initial draft of the paper, E.R.E. participated in sampling and conducted analyses, A.G. conceived the study, conducted analyses and participated in

the interpretation of the results, G.W.L. funded the project, planned and lead the cruise and participated in the interpretation of the results, A.K. participated in the interpretation of the results, M.Y. participated in sampling and conducted analyses. All authors participated in the preparation of the final version of manuscript and have approved it.

## Additional information

**Competing interests:** The authors declare no competing interests.

