## [Peer Review File · Nature Communications]

Reviewers' comments:

Reviewer #1 (Remarks to the Author):

The authors present fluid and mineral speciation data and associated S isotope compositional data sampled from the high temperature vents and the plume directly above vent orifice at the East Pacific Rise 9N hydrothermal system. They highlight the formation mechanism of sulfide minerals in the hydrothermal plume and demonstrate that Fe can persist as nano-pyrite and be transported far distances in the ocean. From my perspective, the manuscript presents a study that is thorough, novel, and an important contribution to the geosciences.

I recommend publication but would like to suggest that the authors better illustrate the particle settling mechanisms/mineral formation processes and how it changes the S isotope composition of the dissolved H₂S and associated particles in the plume. Maybe a “cartoon” in tandem with Figure 4 showing the reaction mechanism and associated S isotope effects, specifically at the different locations sampled in the plume.

Comments and Suggested Edits:

Equation 1a and 1b: Remain consistent with the charge for Fe, for example Fe²⁺ versus Fe(II).

Line 137: delete “to”

Line 139: Syverson et al. (24) did not analyze the S isotope composition of pyrite from the Bio9 vent, rather, Rouxel et al., 2008 is responsible for performing this measurement, see “Integrated Fe- and S-isotope study of seafloor hydrothermal vents at East Pacific Rise 9–10°N”, Chemical Geology.

Line 140: change “the values” to the “the S isotope composition”

Line 140-144: Remain consistent with terminology, such as “light” and “enriched”. I do understand the statement but if someone else read it, it may not be clear what the AVS pool is isotopically enriched in, 34 or 32?

Figure 2: Make sure to specify that the experimentally determined equilibrium fractionation used for pyrite-H₂S is for elevated temperature, 350C. Also, the authors could mention that it is not clear what the S isotope fractionation may be at lower temperature? Is there an S isotope fractionation crossover at lower T? At equilibrium at high T, pyrite is enriched in the light isotopes of S relative to dissolved H₂S at high temperature, which was an unexpected result and in stark contrast to past predictions. Also, it would be helpful if the authors put a representative error bar in the figures to demonstrate the significance of each data point.

Figure 4: The text in the image describing the particle settling is not clear.

Reviewer #2 (Remarks to the Author):

The purpose of this study was to outline processes that control the formation, precipitation and distribution of iron sulfide particles. The authors determined the concentration and speciation of Fe and sulfide as well as the isotopic distribution of sulfide in buoyant plumes from three hydrothermal vents at the East Pacific Rise. The main finding is that while other metal sulfides (including FeS) precipitate within 20 cm from the vent orifice, pyrite nanoparticles persist beyond 100 cm. The authors show that pyrite nanoparticles can represent more than 50% of 'dissolved' Fe thus their stability and persistence suggests they can be an important pathway for the transport of iron to the ocean at large. The study is novel and the conclusions are supported by the data. Overall, I believe the study is interesting and worth of publication in Nature Communications following minor revisions.

1) The main criticism I have of the paper is that the reasons for the persistence of nanoparticulate pyrite despite settling of other metal sulfide particles are not discussed. In the concluding remarks, sentence beginning on line 206, " Here using a combination of we provide a process-based description of how these nanoparticles serve to stabilise Fe'. But what processes stabilize the pyrite nanoparticles and prevent further growth and bulk precipitation? On line 212-214 the authors mention that " The formation of nanoparticulate pyrite and limits on its growth due to mixing act to stabilise the hydrothermally derived Fe", but there is no clear discussion anywhere in the text that links how the processes outlined during mixing limit pyrite nanoparticle growth.

2) Line 19 – 21: Sentence beginning with, “Extensive particle precipitation.....” would be clearer to readers if broken in two sentences or if commas are added.

3) Line 58: “.....and therefore affect the speciation other elements.....” should be ‘of’ other elements.

4) Line 96: Should be referring to Table S1 and not Table S2.

5) Line 104: Equation 1a is not balanced. There should be 4 protons on the left hand side.

6) Line 118-120: Close parenthesis missing and the word ‘after’ should not appear here.
“.....nanoparticulate pyrite (defined operationally asin the $\leq 0.2 \mu\text{m}$ filtered fraction) after” .

7) Line 244 and 92 in the SI: “Minerology as determined from XRD was constrained using elemental results obtained from SEM/EDS”. The word ‘constrained’ does not seem to be the right word to use here.

Reviewers' comments:

Reviewer #1 (Remarks to the Author):

Review:

Iron and sulfide (nano)particle formation and transport in nascent hydrothermal vent plumes

by Alyssa J. Findlay et al.

The authors present fluid and mineral speciation data and associated S isotope compositional data sampled from the high temperature vents and the plume directly above vent orifice at the East Pacific Rise 9N hydrothermal system. They highlight the formation mechanism of sulfide minerals in the hydrothermal plume and demonstrate that Fe can persist as nano-pyrite and be transported far distances in the ocean. From my perspective, the manuscript presents a study that is thorough, novel, and an important contribution to the geosciences.

I recommend publication but would like to suggest that the authors better illustrate the particle settling mechanisms/mineral formation processes and how it changes the S isotope composition of the dissolved H₂S and associated particles in the plume. Maybe a “cartoon” in tandem with Figure 4 showing the reaction mechanism and associated S isotope effects, specifically at the different locations sampled in the plume.

-Drew Syverson

Comments and Suggested Edits:

Equation 1a and 1b: Remain consistent with the charge for Fe, for example Fe²⁺ versus Fe(II).

Fe²⁺ refers to aqueous ferrous Fe, e.g. Fe(H₂O)₆²⁺, whereas Fe(III) refers to ferric iron of unknown speciation. The charges were therefore used as is because it is likely aqueous Fe²⁺ that is oxidised (rather than Fe(II) bound by ligands or as FeS), yet the speciation of the produced oxidised Fe is unknown, and may be Fe oxide nanoparticles.

However, we understand that this may lead to confusion and have changed the Fe(III) charge to Fe³⁺, as this is likely the immediate product of the reaction.

Line 137: delete “to”

This has been done.

Line 139: Syverson et al. (24) did not analyze the S isotope composition of

pyrite from the Bio9 vent, rather, Rouxel et al., 2008 is responsible for performing this measurement, see “Integrated Fe- and S-isotope study of seafloor hydrothermal vents at East Pacific Rise 9–10°N”, Chemical Geology.

Thank you, this reference has been updated.

Line 140: change “the values” to the “the S isotope composition”

This has been done.

Line 140-144: Remain consistent with terminology, such as “light” and “enriched”. I do understand the statement but if someone else read it, it may not be clear what the AVS pool is isotopically enriched in, 34 or 32?

Thank you; “enriched” has been changed to “heavy” to make this clear.

Figure 2: Make sure to specify that the experimentally determined equilibrium fractionation used for pyrite-H₂S is for elevated temperature, 350C. Also, the authors could mention that it is not clear what the S isotope fractionation may be at lower temperature? Is there an S isotope fractionation crossover at lower T? At equilibrium at high T, pyrite is enriched in the light isotopes of S relative to dissolved H₂S at high temperature, which was an unexpected result and in stark contrast to past predictions. Also, it would be helpful if the authors put a representative error bar in the figures to demonstrate the significance of each data point.

A note about the uncertainty regarding isotope fractionation at lower temperatures has been added to the figure legend: “(the *experimental* equilibrium values were taken from Syverson et al.²⁴ and were determined for $T = 350$ °C. We note that it is not certain what the isotope fractionation is at lower temperatures).

Error bars have been added to all points.

Figure 4: The text in the image describing the particle settling is not clear.

Figure 4 has been revised.

Reviewer #2 (Remarks to the Author):

The purpose of this study was to outline processes that control the formation, precipitation and distribution of iron sulfide particles. The authors determined the concentration and speciation of Fe and sulfide as well as the isotopic distribution of sulfide in buoyant plumes from three hydrothermal vents at the East Pacific Rise. The main finding is that while other metal sulfides (including FeS) precipitate within 20 cm from the vent orifice, pyrite nanoparticles persist beyond 100 cm. The authors show that pyrite nanoparticles can represent more than 50% of ‘dissolved’ Fe thus their stability and persistence

suggests they can be an important pathway for the transport of iron to the ocean at large. The study is novel and the conclusions are supported by the data. Overall, I believe the study is interesting and worth of publication in Nature Communications following minor revisions.

1) The main criticism I have of the paper is that the reasons for the persistence of nanoparticulate pyrite despite settling of other metal sulfide particles are not discussed. In the concluding remarks, sentence beginning on line 206, “Here using a combination of we provide a process-based description of how these nanoparticles serve to stabilise Fe’. But what processes stabilize the pyrite nanoparticles and prevent further growth and bulk precipitation? On line 212-214 the authors mention that “The formation of nanoparticulate pyrite and limits on its growth due to mixing act to stabilise the hydrothermally derived Fe, but there is no clear discussion anywhere in the text that links how the processes outlined during mixing limit pyrite nanoparticle growth.

We appreciate that it was not clear in the manuscript which processes were meant to lead to stabilisation of nanoparticulate pyrite. Here, we specifically meant the formation of pyrite nanoparticles through fast nucleation at high temperatures and limits on particle growth due to rapidly falling temperatures during mixing of hydrothermal fluids and seawater, which slows the pyrite formation reaction considerably. We have added text to the manuscript body to state this clearly in the discussion of the temperature dependence of pyrite formation:

“At high temperatures, nucleation rates are high, favouring the formation of nanoparticles²⁷, and the rapidly cooling temperatures and decreasing reactant concentrations during mixing limit further growth, and thereby settling, of the nanoparticles.” (Lines 136 - 139)

We have also added a sentence to the end of the main text:

“Due to their small size¹² and resistance to oxidation³⁶, pyrite nanoparticles may thus mediate Fe export from the vents to the global oceans.” (Lines 198 – 199)

And we have added clarifying text to the conclusions as well:

The formation of nanoparticulate pyrite *at high temperatures* and limits on its growth due to *rapidly cooling temperatures during* mixing act to stabilise the hydrothermally derived Fe as nanoparticulate pyrite and prevent further particle growth and oxidation.” (Lines 210 - 213)

We deleted the text “[...] provide a process-based description of how [...]” to which the reviewer referred, as this detracts from the main point of the sentence, namely that the nanoparticles do persist, despite precipitation of bulk particles. We hope that the processes are now clear from the text added, as detailed above.

2) Line 19 – 21: Sentence beginning with, “Extensive particle precipitation.....”

would be clearer to readers if broken in two sentences or if commas are added.

Commas have been added (shown in red) so that the sentence now reads: "Extensive particle precipitation upon emission of hydrothermal fluids, due to temperature and pH changes during mixing with ambient seawater, controls metal speciation and the magnitude of metal export."

3) Line 58: ".....and therefore affect the speciation other elements....." should be 'of' other elements.

This has been changed.

4) Line 96: Should be referring to Table S1 and not Table S2.

Yes indeed, this has been fixed.

5) Line 104: Equation 1a is not balanced. There should be 4 protons on the left hand side.

This has been fixed.

6) Line 118-120: Close parenthesis missing and the word 'after' should not appear here. ".....nanoparticulate pyrite (defined operationally asin the $\leq 0.2 \mu\text{m}$ filtered fraction) after".

This has been changed.

7) Line 244 and 92 in the SI: "Minerology as determined from XRD was constrained using elemental results obtained from SEM/EDS". The word 'constrained' does not seem to be the right word to use here.

"Constrained" has been changed to "checked compatibility with" in both cases.

REVIEWERS' COMMENTS:

Reviewer #1 (Remarks to the Author):

The authors addressed both reviewers questions, comments, and suggested edits. The manuscript is novel and will provide important information regarding Fe cycling due to mid-ocean ridge hydrothermal processes.

I recommend the manuscript for publication in Nature Communications.

- Dr. Drew D. Syverson

Reviewer #2 (Remarks to the Author):

The changes made by the authors are satisfactory. I recommend publication of the manuscript.